# Unbiased Evaluation of Large Language Models from a Causal Perspective

**Meilin Chen** [* 1]  **Jian Tian** [* 1]  **Liang Ma** [1]  **Di Xie** [1]  **Weijie Chen** [1]  **Jiang Zhu** [1]

## Abstract

Benchmark contamination has become a significant concern in the LLM evaluation community. Previous Agents-as-an-Evaluator address this issue by involving agents in the generation of questions. Despite their success, the biases in Agents-as-an-Evaluator methods remain largely unexplored. In this paper, we present a theoretical formulation of evaluation bias, providing valuable insights into designing unbiased evaluation protocols. Furthermore, we identify two type of bias in Agents-as-an-Evaluator through carefully designed probing tasks on a minimal Agents-as-an-Evaluator setup. To address these issues, we propose the Unbiased Evaluator, an evaluation protocol that delivers a more comprehensive, unbiased, and interpretable assessment of LLMs. Extensive experiments reveal significant room for improvement in current LLMs. Additionally, we demonstrate that the Unbiased Evaluator not only offers strong evidence of benchmark contamination but also provides interpretable evaluation results.

## 1. Introduction

Recently, proprietary models such as GPT-4 (Achiam et al., 2023), Claude(Anthropic, 2024), Gemini(Team et al., 2023), and open-source ones, such as Llama(Touvron et al., 2023), Mistral(Jiang et al., 2023), Qwen(Bai et al., 2023), Yi(AI et al., 2024) have demonstrated remarkable capabilities in natural language processing tasks and beyond. As the capabilities of community models continue to grow, the importance of robust and fair model evaluation becomes increasingly critical. The community has made great efforts to evaluate model performance by expanding the comprehensiveness of benchmarks(Wang, 2018; Wang et al., 2019; Srivastava et al., 2022; Hendrycks et al., 2021; Liang et al.,

---

[*]Equal contribution [1]Hikvision Research Institute. Correspondence to: Meilin Chen <merlinarer@gmail.com>, Liang Ma <maliang6@hikvision.com>.

*Proceedings of the 42^nd International Conference on Machine Learning*, Vancouver, Canada. PMLR 267, 2025. Copyright 2025 by the author(s).

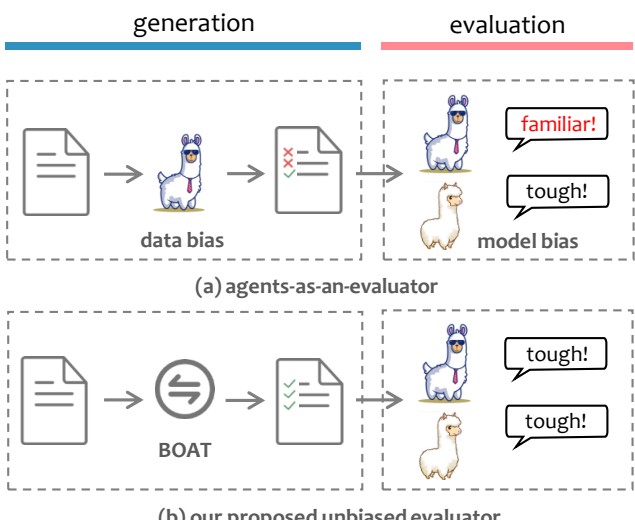

*Figure 1.* (a) Agents-as-an-Evaluator suffers from data and model bias. (b) Our proposed Unbiased Evaluator dynamically evaluate the LLMs with designed *B*ags *O*f *A*tomic In*T*erventions (**BOAT**).

2022; White et al., 2024), or by introducing more complex and challenging tasks to push the boundaries of model capabilities(Wei et al., 2024; He et al., 2024b; Lightman et al., 2023; AI-MO, 2024; 2023; He et al., 2024a).

Despite their success, public benchmarks, most widely used to assess and compare model performance, are particularly vulnerable to contamination issues(Lovin, 2023; Bender et al., 2021; Kocoń et al., 2023; Li, 2023; Zhou et al., 2023; Ni et al., 2024), which are increasingly inevitable due to the scale of training data used in modern models. Recently, agent-based evaluation have been proposed to address contamination issue(Zhu et al., 2024; Liu et al.). Among them, MPA (Zhu et al., 2024) proposes to involve probing and judging agents to automatically transform existing problems in benchmarks into new ones. CogMath(Liu et al.) decouple questions into several evaluation dimensions via an multi-agent system for evaluating LLM's mathematical abilities.

We termed this paradigm as *Agents-as-an-Evaluator*. Formally, Agents-as-an-Evaluator refers to an LLM-based evaluation paradigm in which LLMs (or Agents) not only assess responses but also actively contribute to generating evaluation criteria and questions. Different from previous LLM-

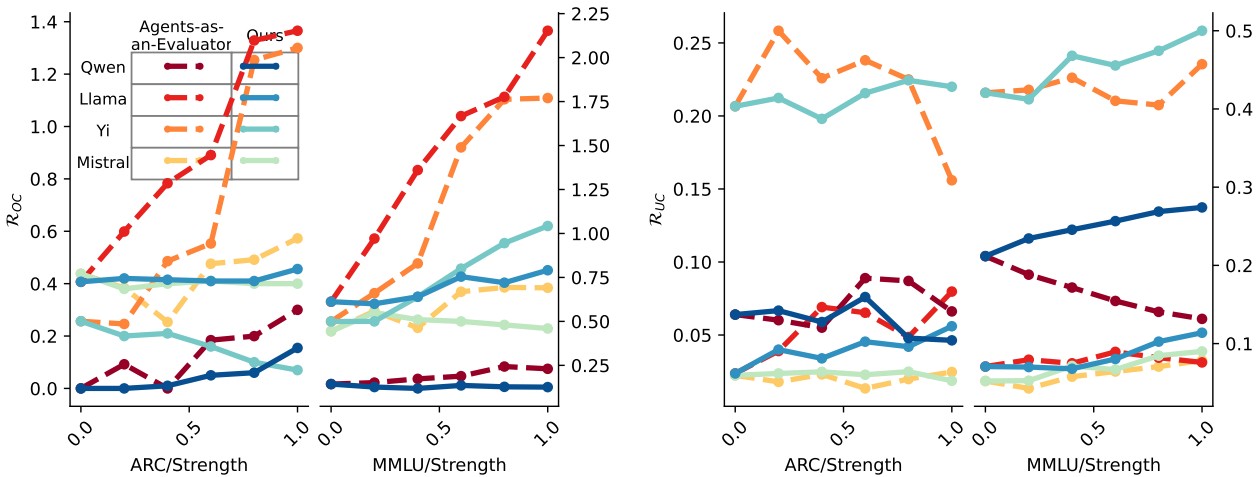

*Figure 2.* Model bias visualizations. Left: $\mathcal{R}_{OC}$ vs. Strength on ARC-C and MMLU datasets. Right: $\mathcal{R}_{UC}$ vs. Strength on ARC-C and MMLU datasets. Strength refers to the probability defined in Equation 3. A higher strength value indicates a greater proportion of "processed" samples within the dataset ("process" denotes rephrasing and BOAT in Agents-as-an-Evaluator and Unbiased Evaluator, respectively). The point where strength=0 represents the original datasets. Mistral, Yi, Llama, Qwen represents Mistral-Large-2411, Yi1.5-34B-Chat, Llama3.1-70B-Instruct and Qwen2.5-72B-Instruct, respectively. For Agents-as-an-Evaluator, we observe a significant increase in $\mathcal{R}_{OC}$ with growing strength, while $\mathcal{R}_{UC}$ remains relatively stable, indicating the existence of model bias. Compared with Agents-as-an-Evaluator, our Unbiased Evaluator remains relatively stable on both $\mathcal{R}_{OC}$ and $\mathcal{R}_{UC}$.

as-a-Judge (Zheng et al., 2023), which operates on the evaluation side by solely determining whether something falls within the scope of a given rule, Agents-as-an-Evaluator extends beyond assessment to the generation side, where LLMs actively contribute to the design of the very questions involved in the task. Considering that prior works have revealed that LLM-as-a-Judge posses certain biases(Blodgett et al., 2020; Ahn et al., 2022; Ferrara, 2023; Gallegos et al., 2024), compared to LLM-as-a-Judge, the critical step of question generation in Agents-as-an-Evaluator may introduce a greater potential for bias. Therefore, an important question rises:

***To what extent are Agents-as-an-Evaluator biased and is there a simple, unbiased alternative for benchmark contamination?***

In this paper, we begin with a theoretical formulation of evaluation bias, followed by an in-depth analysis of the potential bias in designing evaluation protocols. Our results demonstrate that evaluation bias can be decomposed into three components: original, independent and related terms. Therefore, the key takeaway for future research in designing evaluation protocols is that any newly introduced biases should ideally mitigate and, more importantly, counteract existing biases in the original benchmark.

To conduct bias analysis, we design a minimum Agents-as-an-Evaluator, i.e. an LLM is tasked to rephrase the questions without changing their meaning. Build upon this, together with our designed bias probing task, we empirically reveal

that Agents-as-an-Evaluator exhibit two biases: data bias and model bias. As shown in Fig. 1 (a), data bias emerges from accuracy imbalance across different domains during generation, while model bias stems from inherent unfairness during evaluation.

To mitigate the bias in existing methods, we propose the ***Unbiased Evaluator***, as shown in Fig. 1 (b), an evaluation protocol grounded in a causal perspective. Drawing inspiration from interventions in causal inference, where AI systems are tasked with responding to manipulated pairs to understand how altering one factor ( "intervention") impacts other variables in a complex system. Specifically, the evaluation process is formulated as a causal diagram, where LLMs reason over input variables to arrive at the final answer. Taking advantage of this, we design ***B***ags ***O***f ***A***tomic In***T***erventions (**BOAT**) to dynamically evaluate the LLMs. By combining these interventions, the Unbiased Evaluator provides a more comprehensive, unbiased and interpretable assessment.

In summary, our contributions are three-fold: (1) A theoretical formulation of evaluation bias, offering valuable findings for the importance of minimizing the relative term when designing evaluation protocols. (2) The first comprehensive bias analysis for Agents-as-an-Evaluator, revealing data and model bias which undermine the reliability and trustworthiness of Agents-as-an-Evaluator. (3) An unbiased evaluation protocol, Unbiased Evaluator, provides a more comprehensive, unbiased and interpretable assessment for benchmark contamination.

*Table 1.* Data bias. The Pearson and Kendall correlation between domain acc. of MMLU and $\mathcal{R}_{CE}$ across four LLMs.

| Models | Agents-as-an-Evaluator | | | | Ours | | | |
| --- | --- | --- | --- | --- | --- | --- | --- | --- |
| | Kendall | | Pearson | | Kendall | | Pearson | |
| | $\tau$ | $p$-value | $c$ | $p$-value | $\tau$ | $p$-value | $c$ | $p$-value |
| Mistral-Large-2411 | -0.38 | $2.73 \times 10^{-5}$ | -0.47 | $2.18 \times 10^{-4}$ | 0.02 | 0.76 | 0.07 | 0.59 |
| Yi1.5-34B-Chat | -0.17 | $5.86 \times 10^{-3}$ | -0.38 | $3.52 \times 10^{-3}$ | 0.10 | 0.24 | 0.16 | 0.20 |
| Llama3.1-70B-Instruct | -0.39 | $1.81 \times 10^{-5}$ | -0.45 | $4.86 \times 10^{-4}$ | 0.14 | 0.10 | 0.15 | 0.05 |
| Qwen2.5-72B-Instruct | -0.25 | $5.51 \times 10^{-3}$ | -0.41 | $1.44 \times 10^{-3}$ | 0.07 | 0.41 | 0.10 | 0.12 |

## 2. Related Works

### 2.1. LLMs Evaluation

The rapid growth of large language models (LLMs) underscores the need for increasingly robust and fair evaluation methods. Benchmarks offer an effective alternative for model evaluation. The research community has made significant strides in expanding the comprehensiveness of benchmarks (Wang, 2018; Wang et al., 2019; Srivastava et al., 2022; Hendrycks et al., 2021; Liang et al., 2022; White et al., 2024), while also introducing more complex and challenging tasks to push the boundaries of model capabilities (Wei et al., 2024; He et al., 2024b; Lightman et al., 2023; AI-MO, 2024; 2023; He et al., 2024a).

Complementing these evaluation benchmarks, our proposed Unbiased Evaluator introduces an evaluation protocol grounded in a causal perspective, offering a more comprehensive and unbiased assessment.

### 2.2. Benchmark Contamination

Recent research has attached great importance to contamination in LLMs. In particular, (Lee et al., 2021; Sainz et al., 2023; McIntosh et al., 2024; Riddell et al., 2024; Jiang et al., 2024) knowledged that contamination poses significant challenges to the reliability and validity of LLM evaluations. Several research studies (Ni et al., 2024) developed various methods to detect data contamination.

Several works (Fan et al., 2023; Lei et al., 2023; Zhu et al., 2023; 2024; Liu et al.) have been proposed to address the contamination issue. Among these, protocols like (Fan et al., 2023; Zhu et al., 2023; Liu et al.) are specifically designed for mathematical tasks, while (Lei et al., 2023) focuses on long-context evaluation. Among the research of Agents-as-an-Evaluator(Zhu et al., 2024; Liu et al.), MPA (Zhu et al., 2024) proposes to involve paraphrasing and judging agents to automatically transform existing problems in benchmarks into new ones. CogMath(Liu et al.) decouple questions into several evaluation dimensions via an multi-agent system for

evaluating LLM's mathematical abilities.

Our proposed Unbiased Evaluator stands in contrast to these, as it is designed to be generalized for a wide range of tasks and ensures an unbiased evaluation.

### 2.3. Evaluation Bias

Recent studies have highlighted that LLM-as-a-Judge exhibit various types of biases across various tasks(Dai et al., 2024; Gallegos et al., 2024; Chen et al.; Ye et al., 2024), such as position bias, length bias, self-enhancement bias etc. These internal biases of LLMs may also affect LLM-as-a-judge, leading to unfair evaluation outcomes and subsequently impacting the development of LLMs.

Unlike prior research that mainly focuses on biases in LLM-as-a-Judge, this paper addresses the biases inherent in the generation of Agents-as-an-Evaluator, an area largely unexplored but critical to understanding the fairness and impact of LLMs in evaluative roles.

## 3. Bias Analysis

### 3.1. Theoretical Analysis

**Formulation of Evaluation Bias.** Consider in an LLM evaluation, where the true capability of a model is parameterized by $\phi$, which is a fixed but unknown quantity we aim to estimate. Given evaluation data $\mathcal{X} = \{x_1, x_2, \ldots, x_n\}$, the estimator $\hat{\phi} = \mathcal{E}(\mathcal{X})$ is a function of the data and provides an approximation of $\phi$. Then, the difference between expectation of $\hat{\phi}$ and $\phi$ can be defined as evaluation bias:

$$\epsilon(\hat{\phi}) = \mathbb{E}[\hat{\phi}] - \phi \tag{1}$$

where $\mathbb{E}[\hat{\phi}]$ is the expected value of the estimator over $\mathcal{X}$. A bias greater than 0 indicates that the estimator overestimates $\phi$, often due to contaminated data. A bias of 0 means the estimator is unbiased, providing a perfect estimate. A bias less than 0 suggests the estimator underestimates $\phi$, which can occur when a model is evaluated on a limited dataset that fails to capture the task's complexity.

**Proposition 3.1.** (*Proof in Appendix D.1*) Given a new designed evaluation protocol, which transforms original benchmark $D$ into $D'$. Then, the strength of bias with the new evaluation protocol can be decomposed into three terms: original, related and independent term.

$$\mathbb{E}[\epsilon(\hat{\phi}_{D'})^2] = \underbrace{\mathbb{E}[\epsilon(\hat{\phi}_D)^2]}_{\text{original}} + \underbrace{2\text{Cov}(\epsilon(\hat{\phi}_D), \Delta)}_{\text{related}}$$
$$+ \underbrace{2\mathbb{E}[\epsilon(\hat{\phi}_D)]\mathbb{E}[\Delta] + \mathbb{E}[\Delta^2]}_{\text{independent}} \quad (2)$$

where, $\Delta$ is the delta bias which arises from the introduction of new evaluation protocol.

- $\mathbb{E}[\epsilon(\hat{\phi}_D)^2]$ is an original term that is associated with the original bias existing in the original benchmark $D$.

- $2\text{Cov}(\epsilon(\hat{\phi}_D), \Delta)$ is a related term which pertains to biases that newly introduced biases are correlated with the pre-existing biases in the original benchmark.

- $2\mathbb{E}[\epsilon(\hat{\phi}_D)]\mathbb{E}[\Delta] + \mathbb{E}[\Delta^2]$ is an independent term, stemming from biases inherent to the methodology itself.

Proposition 3.1 offers a pivotal perspective for future research in designing evaluation protocols that any newly introduced biases should ideally mitigate and, more importantly, counteract existing biases in the original benchmark.

### 3.2. Bias in Agents-as-an-Evaluator

In addition to our theoretical analysis, we conduct extensive experiments in this section to demonstrate that Agents-as-an-Evaluator exhibit two types of bias: data bias and model bias.

- Data bias arises from an imbalance in accuracy across different domains during generation. For example, in tasks involving diverse domains, such as various subjects in MMLU, LLMs tend to excel in domains where they already perform well while struggling significantly in domains where their performance is weaker.

- Model bias originates from inherent unfairness during evaluation. an LLM tends to generate content that aligns more with its implicit strengths, giving it an unfair advantage.

**Discussion.** Previous works(Chen et al.; Ye et al., 2024) have revealed the self-enhanced bias in LLM-as-a-Judge, i.e. LLM judges may favor the answers generated by themselves. Model bias, however, focus more on the bias of generation side in Agents-as-an-Evaluator, which still remain unexplored.

### 3.3. Probing Task

In this section, we design a probing task to detect the potential bias in Agents-as-an-Evaluator.

**Minimum Agents-as-an-Evaluator.** To perform bias analysis, following (Zhu et al., 2024), we design a minimal Agents-as-an-Evaluator framework. Specifically, given an original benchmark $\mathcal{D}$, an LLM is tasked with rephrasing the questions in $\mathcal{D}$ without altering their meaning. Formally, this process generates a new benchmark, $\mathcal{D}' = LLM(\mathcal{D})$.

**Task Design.** Given an original benchmark $\mathcal{D}$ on with $m$ evaluation problems $\{x_1, x_2, \ldots, x_m\}$, and $n$ cutting edged LLMs $\mathcal{M} = \{\theta_1, \theta_2, \ldots, \theta_n\}$, $\mathcal{D}'_i$ denotes the rephrased benchmark using $i$-th LLM $\theta_i$, rephrased $j$-th evaluation problem $x'_{ij}$ in $\mathcal{D}'_i$ is rephrased from $d_j$ with probability $p$, i.e.

$$\mathcal{D}'_i = \{x'_{i1}, x'_{i2}, \ldots, x'_{im}\}$$
$$x'_{ij} = \begin{cases} \theta_i(x_j) & \texttt{random} \le p \\ x_j & \texttt{random} > p \end{cases} \quad \forall x'_{ij} \in \mathcal{D}'_i \quad (3)$$

Note that during probing task, the ground truth label of each problem remains unchanged. With $\mathcal{D}'_i$, we perform LLM-as-a-Judge using all models in $\mathcal{M}$. Specifically, we ask each model to assess its confidence that it thinks the ground truth label is still the right answer for the rephrased question, i.e.

$$\mathcal{S}_{ijk} = \theta_k(x'_{ij})$$
$$\forall i \in [1, n]; \forall j \in [1, m]; \forall k \in [1, n] \quad (4)$$

where $\mathcal{S}_{ijk}$ is the confidence score assessed by $\theta_k$ for question $x'_{ij}$. The score $\mathcal{S}_{ijk}$ lies within the range $[1, 10]$, where a higher score indicates that the model has greater confidence in the ground truth label being the correct answer, while a lower score suggests stronger confidence that the ground truth label is not the correct answer. Detailed prompt is presented in Appendix B.

**Metric design.** To quantify bias during LLMs evaluation, we draw inspiration from group consensus in human society, where collective opinions are often regarded as more reliable and fair compared to individual perspectives. Building on this idea, we propose three metrics: $\mathcal{R}_{CE}$(Consensus-Error Rate), $\mathcal{R}_{OC}$(Over-Confidence Rate) and $\mathcal{R}_{UC}$(Under-Confidence Rate), incorporating both self and collective judgment.

$$\mathcal{R}_{CE} = \frac{1}{m} \sum_{j=1}^{m} [\underbrace{(10 - \mathcal{S}_{iji}) * (10 - \hat{\mathcal{S}}_{ij})}_{\substack{\text{consensus} \\ \text{intensity}}}] * \underbrace{\prod_{k=1}^{n} sgn(\mathcal{S}_{ijk} < tu)}_{\substack{\text{all consensus}}}$$
$$(5)$$

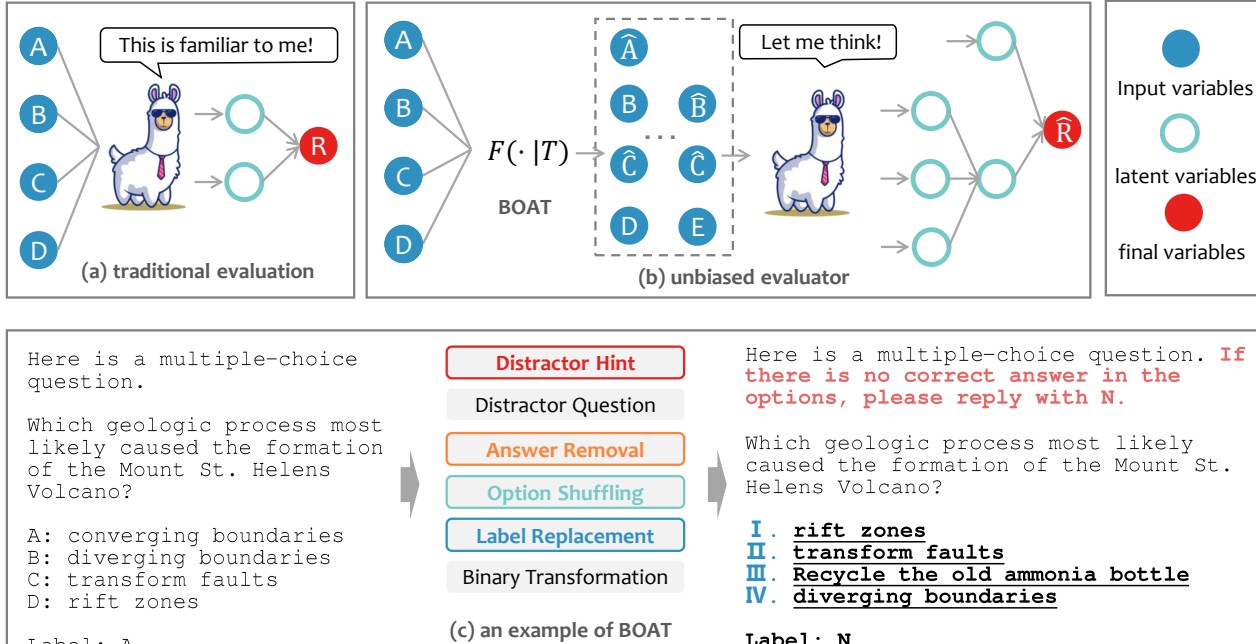

*Figure 3.* (a) Traditional evaluation methods rely on static and fixed variables, suffering from contamination issues. (b) Unbiased Evaluator enhance the evaluation process by augmenting these variables through carefully designed ***B***ags ***O***f ***A***tomic In***T***erventions (**BOAT**). (c) An example of BOAT. Underlined contents are derived from multiple interventions.

where *sgn* is a sign function. $tu$ is a upper threshold of "NO". In this paper, following prompt in probing task, $tu$ is set to 5. $\hat{\mathcal{S}}_{ij}$ is the averaged score among the other models: $\hat{\mathcal{S}}_{ij} = \frac{1}{n-1} \sum_{k=1 \; k!=i}^{n} (\mathcal{S}_{ijk})$.

$$\mathcal{R}_{OC} = \frac{1}{m} \sum_{j=1}^{m} [\; \underbrace{\mathcal{S}_{iji} * (10 - \hat{\mathcal{S}}_{ij})}_{\text{conflict intensity}} \;]$$

$$* \underbrace{sgn\,[\mathcal{S}_{iji} > tl]}_{\text{self confidence}} * \underbrace{\prod_{k=1 \; k!=i}^{n} sgn(\mathcal{S}_{ijk} < tu)}_{\text{collective consensus}} \quad (6)$$

$$\mathcal{R}_{UC} = \frac{1}{m} \sum_{j=1}^{m} [\; \underbrace{(10 - \mathcal{S}_{iji}) * \hat{\mathcal{S}}_{ij}}_{\text{conflict intensity}} \;]$$

$$* \underbrace{sgn\,[\mathcal{S}_{iji} < tu]}_{\text{self confidence}} * \underbrace{\prod_{k=1 \; k!=i}^{n} sgn(\mathcal{S}_{ijk} > tl)}_{\text{collective consensus}} \quad (7)$$

where $tl$ is lower threshold of "Yes", and $tl$ is set to 6.

The Consensus-Error Rate $\mathcal{R}_{CE}$ quantifies the overall consensus in predicting "NO", with a weight that increases as the confidence values $\mathcal{S}_{iji}$ and $\hat{\mathcal{S}}_{ij}$ decrease, reflecting higher confidence in predicting "NO". On the other

hand, the Over-Confidence Rate $\mathcal{R}_{OC}$ measures the scenario where a LLM exhibits high self-confidence in predicting "Yes", while the collective consensus predicts "NO". In contrast, the Under-Confidence Rate $\mathcal{R}_{UC}$ quantifies the scenario where a LLM exhibits high confidence in predicting "NO", while the collective consensus predicts "Yes".

**Data Bias.** We treat each subject in MMLU as a domain and calculate the Pearson and Kendall correlation between domain accuracy and $\mathcal{R}_{CE}$ across four LLMs. The results, presented in Table 1, reveal a relatively strong negative correlation with small $p$-values. This indicates that LLMs tend to perform better in domains with lower $\mathcal{R}_{CE}$ values while facing greater challenges in domains where their performance is weaker.

**Model Bias.** In Fig. 2, we visualize how $\mathcal{R}_{OC}$ and $\mathcal{R}_{UC}$ evolve as the strength parameter $p$ in Eq. (4) increases. We observe a significant increase in $\mathcal{R}_{OC}$ with growing strength, particularly for models like Llama3.1-70b-Instruct and Yi-34B-Chat. In contrast, $\mathcal{R}_{UC}$ remains relatively stable. This suggests that transitioning from the original evaluation protocol ($p = 0$) to the Agents-as-Evaluator ($p = 1$) causes LLMs to generate content that aligns closer with their implicit strengths ($\mathcal{R}_{OC}$), rather than diverging from them ($\mathcal{R}_{UC}$), ultimately providing themselves with an unfair advantage.

# 4. Unbiased Evaluator

## 4.1. General formulation

Consider evaluating an LLM $\theta$, where the model is tasked with generating answers for a set of evaluation problems $\mathcal{X} = \{x_1, x_2, \ldots, x_n\}$. The ground truth labels $\mathcal{Y} = \{y_1, y_2, \ldots, y_n\}$ are then used to calculate performance metrics. For each problem $x_i \in \mathcal{X}$ and its corresponding label $y_i \in \mathcal{Y}$, the evaluation process can be framed as a causal analysis, where the input $\mathcal{X}$ and the output $\mathcal{Y}$ represent the cause and effect, respectively.

The evaluation process for a question $x$ can generally be represented as a causal diagram, specifically a causal directed acyclic graph (DAG) $\mathcal{G} = (E, V)$. In this representation, the nodes or vertices $V = \{V_{\text{input}}, V_{\text{latent}}, R\}$ correspond to the observed input variables $V_{\text{input}}$, unobserved latent variables $V_{\text{latent}}$, and the ground truth label $R$. The edges $E$ capture the direct causal relationships between these variables.

Specifically, $V_{\text{input}} = \{A, B, C, D\}$ represents the input variables, which are composed of several atomic elements derived from the input samples. In widely used multiple-choice questions, atomic variables may include elements such as the instruction, answer labels, and other context-specific components. For instance, consider the question: "Is 9.8 bigger than 9.11? A: True, B: False". Here, the atomic variables include "9.8", "bigger", "9.11", "A", "B", "True", and "False". A large language model (LLM) reasons over these atomic variables to arrive at the final answer.

Traditional evaluation methods, as depicted in Figure 3 (a), rely on static and fixed inputs, i.e., input variables, to investigate causal effects. However, this rigid framework may inadvertently introduce contamination issues, limiting the validity and robustness of the causal analysis. In contrast, our proposed approach, termed the Unbiased Evaluator, adopts a more dynamic and adaptive strategy, as shown in Figure 3 (b). Specifically, given input variables and ground truth label pairs $\{\{A, B, C, D\}, R\}$ for a task $\mathcal{T}$, we enhance the evaluation by augmenting variables through carefully designed **B**ags **O**f **A**tomic In**T**erventions (**BOAT**), i.e. new input variable configurations, such as $\{\{\hat{A}, B, \hat{C}, D\}, \hat{R}\}$ and $\{\{\hat{B}, \hat{C}, E\}, \hat{R}\}$, are intervened with $\mathcal{F}(\cdot|\mathcal{T})$. More intervened examples are presented in Appendix E.

Our Unbiased Evaluator aims to assess whether models can genuinely answer a question correctly by employing causal interventions that align with human recognition. Unbiased Evaluator not only mitigates potential contamination by incorporating diverse interventions but also offers an interpretable framework for evaluating LLMs. By examining counterfactual scenarios, such as comparing the effects of $\hat{A}$ versus $A$, our method fosters a deeper and more transparent understanding of model performance. We demonstrate this in the following experiments procedures.

## 4.2. Bags of Atomic Interventions

Based on the general formulation described above, we provide a demo in Table 6 for a clear demonstration of our designed Bags of Atomic Interventions. Specifically, we focus on two widely used tasks: Multiple Choice Questions (MCQ) and Mathematics. For Multiple Choice Questions, we have designed six atomic interventions targeting different intervention positions within the task.

- **Distractor Hint.** A hint is introduced in the form of an additional option indicating that no correct answer exists for the question.

- **Distractor Question.** To ensure the model thoroughly understands the question, a distractor question is introduced, randomly selected from the same dataset.

- **Answer Removal.** Building upon the first strategy, some answers (including correct or not) is removed and replaced with an unrelated option from other questions.

- **Option Shuffling.** The order of the options is randomly shuffled to investigate whether the model's performance is influenced by their positional arrangement.

- **Label Replacement.** The conventional option labels (*A, B, C, D*) are replaced with numerical labels (*1, 2, 3, 4* or *I, II, III, IV*) to explore whether the labeling format affects the model's output tendencies.

- **Binary Transformation.** The stem of a multiple-choice question is combined with each of the four options to create four true/false questions.

For mathematics, we design two interventions to transform questions into True/False question and MCQ, and then combined with aforementioned MCQ interventions.

- **Question Jitter.** Slightly altering the numbers in a question allows an open-ended math problem to be reframed as a True/False question.

- **Answer Jitter.** To transform an open-ended math problem into a multiple-choice question, the numerical answer can be adjusted with minor variations.

Note that these atomic interventions can be organically combined to form complex interventions, providing a more comprehensive evaluation of large language models. For instance, Distractor Hint can be seamlessly integrated with Answer Removal, Option Shuffling, and Label Replacement to create a more sophisticated and targeted intervention.

**Discussion.** The design of BOAT is guided by the findings from Proposition 3.1. Specifically, the mechanisms of Answer Removal and Binary Transformation are introduced to

*Table 2.* The performance of different LLMs on vanilla benchmarks and Unbiased Evaluator.

| Model | ARC-C | | | MMLU | | | GSM8K | | |
|---|---|---|---|---|---|---|---|---|---|
| | Vanilla | Ours | Δ | Vanilla | Ours | Δ | Vanilla | Ours | Δ |
| GPT-4o | 94.51 | 89.36 ± 0.26 | 5.15 | 83.51 | 68.82 ± 0.42 | 14.69 | 96.21 | 86.01 ± 0.64 | 10.20 |
| GPT-4-Turbo | 96.48 | 89.74 ± 0.13 | 6.74 | 84.10 | 71.81 ± 0.28 | 12.29 | 97.73 | 90.18 ± 0.34 | 7.55 |
| Gemini-2.0 | 95.71 | 88.15 ± 0.26 | 7.56 | 86.20 | 73.57 ± 0.09 | 12.63 | 97.88 | 89.35 ± 0.04 | 8.53 |
| Qwen2.5-72B-Instruct | 94.33 | 85.69 ± 0.86 | 8.64 | 84.07 | 69.56 ± 0.39 | 14.51 | 98.41 | 88.86 ± 0.89 | 9.55 |
| Llama3.1-70B-Instruct | 93.99 | 81.23 ± 0.74 | 12.76 | 80.70 | 61.93 ± 0.12 | 18.77 | 95.98 | 82.97 ± 0.66 | 13.01 |
| Yi1.5-34B-Chat | 93.91 | 71.79 ± 0.45 | 22.12 | 77.72 | 56.70 ± 0.18 | 21.02 | 91.96 | 69.60 ± 0.92 | 22.36 |
| Mistral-Large-2411 | 94.85 | 81.89 ± 0.31 | 12.96 | 82.37 | 61.63 ± 0.09 | 20.74 | 97.73 | 90.04 ± 0.68 | 7.69 |

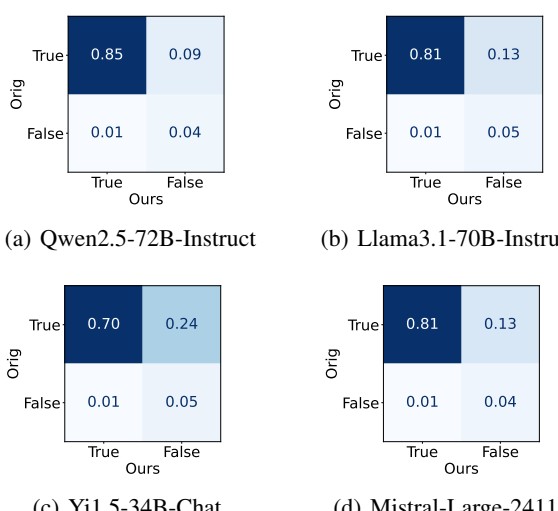

(a) Qwen2.5-72B-Instruct          (b) Llama3.1-70B-Instruct

(c) Yi1.5-34B-Chat          (d) Mistral-Large-2411

*Figure 4.* The confusion matrix of original benchmarks and Unbiased Evaluator on ARC-C.

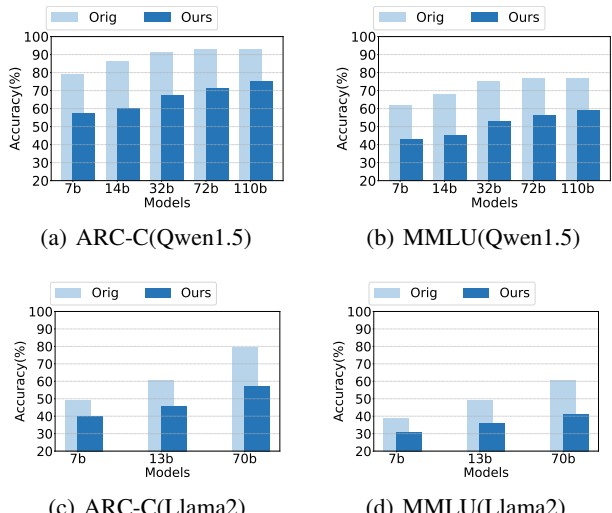

(a) ARC-C(Qwen1.5)          (b) MMLU(Qwen1.5)

(c) ARC-C(Llama2)          (d) MMLU(Llama2)

*Figure 5.* The performance of Qwen1.5 series and Llama2 series models on ARC-C and MMLU, considering both original benchmarks and Unbiased Evaluator.

address biases inherent in the original benchmark, thereby mitigating related term. For instance, public benchmarks like MMLU are known to exhibit ambiguities, which will be alleviated with these interventions. Additionally, we have made effort to minimize independent term introduced by our method (refer to the implementation details).

## 5. Experiments

### 5.1. Experimental Setup

**Evaluated Datasets and LLMs.** Following (Zhu et al., 2024), we evaluate on two widely used benchmarks for multiple-choice questions: ARC-Challenge (ARC-C) (Clark et al., 2018) and MMLU (Hendrycks et al., 2021). For mathematical problem-solving, we utilize the GSM8K dataset (Cobbe et al., 2021). The evaluation includes three proprietary large language models (LLMs): GPT-4o (OpenAI, 2024), GPT-4-Turbo (Achiam et al., 2023), and Gemini 2.0

(Team et al., 2023). Additionally, we assess several open-source models, including Llama (Touvron et al., 2023), Mistral (Jiang et al., 2023), Qwen (Bai et al., 2023), and Yi (AI et al., 2024).

**Implementation Details.** We build our method on the widely-used OpenCompass (Contributors, 2023) evaluation framework. Following the approach in (Zhu et al., 2024), we set the generation temperature to 0 for all models and cap the output length at a maximum of 1000 tokens. All evaluations are conducted in a 5-shot setting, with results averaged over 5 runs. Moreover, the interventions in BOAT are not randomly combined but follow specific constraints. We regulate the probability of each intervention to ensure balance. When applying a binary transformation to questions, modifications involving phrases such as "which" or "following" were excluded. Furthermore, during the Answer Removal process, we ensured that the answers extracted

from different questions were not identical. For additional details, please refer to Appendix C.

## 5.2. Main Results

Table 2 presents the evaluation results of different LLMs on the original protocol and our Unbiased Evaluator, showing all LLMs experienced performance degradation on the Unbiased Evaluator. Specifically, GPT-4-Turbo and Gemini 2.0 demonstrated the smallest relative performance drop, maintaining their position as the strongest models. In contrast, GPT-4o performed similarly to GPT-4-Turbo on the original protocol but exhibited a larger average performance decline, particularly on MMLU and GSM8K. We hypothesize that the omni design of GPT-4o may hinder its performance on NLP tasks. Compared to proprietary models, open-source ones showed a more significant average drop, with Yi1.5-34B-Chat standing out as particularly affected, suggesting a substantial potential for improvement.

# 6. Ablations

## 6.1. Bias Analysis

Following Sec. 3.3, we conduct an analysis of data and model bias for our proposed method, with the results presented in Table 1 and Fig. 2. Regarding data bias, our method exhibits a smaller coefficient and a large $p$-value, indicating reduced data bias. In terms of model bias, compared to Agents-as-Evaluator, our Unbiased Evaluator remains relatively stable in both $\mathcal{R}_{OC}$ and $\mathcal{R}_{UC}$ as the strength increases, suggesting lower model bias.

## 6.2. Contamination Analysis

Our Unbiased Evaluator aims to assess whether models can genuinely answer a question correctly by employing causal interventions that align with human recognition. Therefore, Unbiased Evaluator provides a more accurate measure of a model's true and robust performance on a given benchmark by eliminating performance inflation caused by data contamination. The decline of accuracy in Table 2 actually reflects the decrease of contamination. To validate this, following (Yang et al., 2023), we further provide an additional fine-tuning ablation study. Specifically, we fine-tune Llama2-13B on the original samples from the MMLU test set and evaluate it on MMLU test set under two conditions: with and without our Unbiased Evaluator.

Results in 3 highlight that our Unbiased Evaluator provides a more rigorous assessment of benchmark contamination. Even when trained directly on the original test set, the model struggles to perform well under the Unbiased Evaluator, suggesting that it effectively mitigates data contamination and ensures a more robust evaluation.

*Table 3.* Contamination analysis. We fine-tune Llama2-13B on the original samples from the MMLU test set and evaluate it on MMLU test set under two conditions: with and without our Unbiased Evaluator.

| train set | w/o | w/ |
|---|---|---|
| Llama2-13B | 55.6 | 33.7 |
| Llama2-13B + original test set | 96.6 | 37.1 |

## 6.3. Evaluation Reliability

Our Unbiased Evaluator demonstrates a significantly stronger alignment with human expert judgments. Since obtaining comprehensive expert evaluations across multiple models is both costly and impractical, we instead benchmark our method against LiveBench (White et al., 2025), a continuously updated and widely recognized evaluation platform. Specifically, we compute both Pearson and Kendall correlation coefficients between our averaged results of Table 2 and the global average scores reported in the latest LiveBench release (2024-11-25). To ensure a fair comparison, we exclude two models—GPT-4-Turbo and Yi1.5-34B-Chat—as they are not covered in the corresponding LiveBench update.

*Table 4.* Evaluation Reliability. We compute both Pearson and Kendall correlation coefficients between our averaged results of Table 2 and the global average scores reported in the latest LiveBench release.

| | Pearson | Kendall |
|---|---|---|
| Vanilla | 0.918 | 0.600 |
| Unbiased Evaluator | 0.949 | 1.000 |

Results in Table 4 confirm that our method aligns more closely with LiveBench. Notably, it achieves a perfect ranking correlation with LiveBench (as measured by Kendall), a significant improvement over baseline. Unlike LiveBench, which covers diverse tasks and requires substantial resources to update questions regularly, ours leverages existing benchmarks and requires almost no additional resources.

## 6.4. Confusion Matrix Analysis

We present a comprehensive confusion matrix in Fig. 4, which evaluates performance across four dimensions: "True/False" labels from the original protocol and our Unbiased Evaluator. For instance, the category labeled "Original True" and "Unbiased Evaluator False" highlights scenarios where the model performs correctly under the original protocol but fails when assessed with the Unbiased Evaluator. A striking observation across all models is the high frequency of "Original True" and "Unbiased Evaluator False"

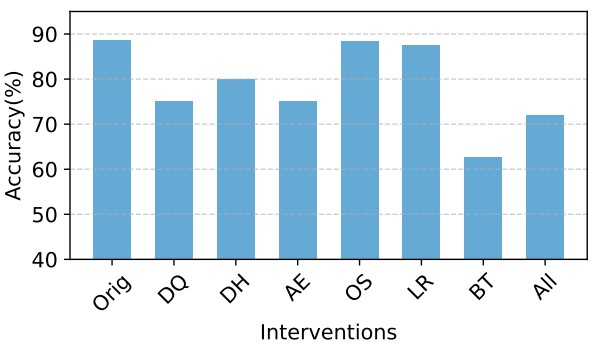

*Figure 6.* Accuracy of Qwen2.5-7B-Instruct on ARC-C when each single atomic interventions are applied. Orig denotes the original benchmark. DQ, DH, AE, OS, LR and BT represents Distractor Hint, Distractor Question, Answer Removal, Option Shuffling, Label Replacement and Binary Transformation, respectively.

instances. This trend suggests issues with data contamination and inherent capability limitations within the models. Notably, the "Original True" and "Unbiased Evaluator False" of Yi1.5-34B-Chat are significantly higher compared to other larger models. This indicates that it may be more susceptible to data contamination and capability constraints.

### 6.5. Effect of Interventions & Evaluation Interpretability

Figure 6 illustrates the accuracy of Qwen2.5-7B-Instruct when each single atomic interventions are applied. The results show that the model's performance varies significantly depending on the type of intervention. Notably, compared to the original protocol, the Binary Transformation intervention poses the greatest challenge, leading to a substantial decline in accuracy. This suggests that the model may not fully comprehend every option in a MCQ but instead might rely on certain heuristic shortcuts to arrive at an answer. Moreover, even with relatively simple interventions, such as Option Shuffling and Label Replacement, the model's performance exhibits a slight degradation. This provides strong evidence of potential data contamination.

### 6.6. Effect of the Scaling

As shown in Figure 5, we evaluate the Qwen1.5 and Llama2 series models with Unbiased Evaluator on MMLU and ARC-C. The results indicate that as the model parameters increase, performance gradually improves. Notably, under the original evaluation protocol, the larger Qwen1.5 models achieved over 90% performance, nearing saturation and limiting the ability to evaluate larger models, such as 32/72/110B. In contrast, Unbiased Evaluator demonstrate a consistent performance improvement from 32B to 110B.

### 6.7. Human Verification

To ensure the correctness of proposed method, we randomly selected 300 samples from the MMLU and 100 samples from the ARC-C and GSM8K, 500 questions in total. 9 human experts (with bachelor or higher degree) are divided into 3 groups, each with 3 person. They were asked to judge the following question: Whether the answers to the intervened questions are correct. As shown in Table 6.7, the human verification demonstrates an overall accuracy rate of 99.3%, 99.9% and 99.7% for ARC-C, MMLU and GSM8K, respectively, indicating the effectiveness of our methodology.

*Table 5.* Results of human verification on ARC-C, MMLU and GSM8K datasets.

|  | ARC-C | MMLU | GSM8K | Average |
|---|---|---|---|---|
| Group 1 | 1.000 | 0.997 | 0.990 | 0.996 |
| Group 2 | 0.990 | 1.000 | 1.000 | 0.997 |
| Group 3 | 0.990 | 1.000 | 1.000 | 0.997 |
| Average | 0.993 | 0.999 | 0.997 | / |

## 7. Future Work

Our proposed method can be easily extended towards open-ended tasks. Most tasks (e.g., multiple-choice, math), as demonstrated in this paper, inherently follow natural rules in either the questions or answers, and rule-based interventions can be automatically applied. The other small percentage of tasks, can use a debiased Agents-as-Evaluator version. Concretely, our study has revealed the data and model biases of previous version, inspiring two designs to mitigate them: (1) **Cross-generation**: to reduce model bias, we can break down question generation into multiple chunks, using different models for each. (2) **Cross-checking**: multiple advanced models can be used to cross-check the output to mitigate data bias and enhance quality. Overall, our method is easily scaled to most tasks, and our insights will provide valuable inspiration for future advancements in evaluation methodologies. We leave these directions to future work.

## 8. Conclusions

This paper present a theoretical analysis of evaluation bias, offering valuable findings for designing protocols. Moreover, two types of bias in Agents-as-an-Evaluator are identified with probing task. To mitigate the bias, guided with previous findings, an new evaluation protocol, Unbiased Evaluator, is proposed to offer unbiased and interpretable assessment for benchmark contamination. We look forward that our method may bring inspirations for future design for LLMs evaluation.

## Impact Statement

Bias in the evaluation of LLMs is a crucial issue for ensuring responsible AI development in society. This work conduct a detailed bias analysis on previous protocol, followed by a novel evaluation protocol designed to fairly measure the true capabilities of LLMs. By providing a more precise assessment framework, this protocol aims to enhance our understanding of these models, ultimately contributing to more transparent, fair, and reliable AI systems.

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

## A. Demo Case of BOAT

*Table 6.* Bags of Atomic Interventions (BOAT). This is a demo case for a clear demonstration. For further details, please refer to Sec. 5.1. For each intervention, we present a simple question to show its effect. The "intervened" column only displays the part that has changed, highlighted in the same color as the original part it replaces. The other parts of the content remain unchanged.

| $\mathcal{T}$ | positions | interventions | examples | | label |
|---|---|---|---|---|---|
| | | | original | intervened | |
| MCQ | instruction | Distractor Hint | Here is a multiple choice question, | Here is a multiple choice question, answer A or B, if there is no answer, reply N. | A |
| | question | Distractor Question | answer A or B. | Is 9.8 bigger than 9.11? 1+1=? | A |
| | answer | Answer Removal Option Shuffling Label Replacement | Is 9.8 bigger than 9.11? A: True B: False | A: 2 B: False A: False B: True I: True II: False | N B I |
| | overall | Binary Transformation | | Here is a multiple choice question, judge True (T) or False (F) for each choice. | TF |
| Math | overall | Question Jitter | Answer this question. 1+1=? | Judge True (T) or False (F) for given possible answer. 1+2=2? | F |
| | overall | Answer Jitter | | Here is a multiple choice question. 1+1=? A: 0 B: 1 C: 2 D: 3 | C |

## B. Prompt in Probing Task

```
I have a question with a possible answer.

### Question:
{question}

### Possible Answer:
{answer}

You need to rate the given possible answer on a scale of 1 to 10 based on the confidence
    of its correctness for the question, using the following rating rules:

Score 1 to 2: You are very confident that the given possible answer is completely
    incorrect for the question.
Score 3 to 4: You are fairly confident that the given possible answer is incorrect, but
    there is a small chance it could be partially correct.
Score 5 to 6: You are uncertain about the correctness of the given possible answer; it
    could be right or wrong.
Score 7 to 8: You are fairly confident that the given possible answer is correct, but
    there is a small chance it could be partially incorrect.
Score 9 to 10: You are very confident that the given possible answer is completely correct
     for the question.

Finally, you must rate the answer strictly on a scale of 1 to 10 with in the format of
    ''<<< >>>'', for example, ''<<<5>>>.''
```

*Table 7.* The BOAT constraints are represented in a matrix where each cell indicates whether the second intervention (column) can be applied when the first intervention (row) is introduced. A check mark (✓) denotes that the second intervention can be combined with the first one, while a black triangle (▲) indicates that the second intervention is a required addition. DQ, DH, AE, OS, LR and BT represents Distractor Hint, Distractor Question, Answer Removal, Option Shuffling, Label Replacement and Binary Transformation, respectively.

| First/Second Intervention | DH | DQ | AR | OS | LR | BT |
|---|---|---|---|---|---|---|
| DH | | ✓ | ✓ | ✓ | ✓ | |
| DQ | ✓ | | | ✓ | ✓ | |
| AR | ▲ | | | ✓ | ✓ | |
| OS | ✓ | ✓ | ✓ | | ✓ | |
| LR | ✓ | ✓ | ✓ | ✓ | | |
| BT | | | | | | |

## C. Implementation Details

### C.1. Hyperparameter

To ensure a balance of interventions in BOAT, the probability for each intervention was set to 0.5, except for Binary Transformation, which was assigned a probability of 0.1. Under the 5-shot evaluation setting, we introduced the same combinations of atomic interventions to the few-shot samples as those in the final question. An exception was made for questions affected by Answer Removal: in such cases, we randomly selected half of the few-shot samples for intervention to prevent the model from repeating the output corresponding to option N.

### C.2. BOAT constraints

As shown in 7, there are constraints among different interventions, and not all interventions can be combined arbitrarily. For example, the Answer Removal can only be introduced when the Distractor Hint intervention exists. Moreover, the binary transformation intervention cannot be combined with other interventions.

Please note that the instructions will change depending on the interventions. The detailed instructions are presented in 8.

*Table 8.* The detailed instructions under different interventions. DQ, DH, AE, OS, LR and BT represents Distractor Hint, Distractor Question, Answer Removal, Option Shuffling, Label Replacement and Binary Transformation, respectively.

| Intervention | Instruction |
|---|---|
| DH | If there is no correct answer in the options, please reply with N. |
| DQ | Here are two questions and only one of them corresponds to the options. Please select the correct answer. |
| AR | Here is a multiple choice question. |
| OS | Here is a multiple choice question. |
| LR | Here is a multiple choice question. |
| BT | The following are true/false questions. If the answer is correct, please reply with T, otherwise reply with F. |

## D. Proof of Proposition

### D.1. Proof of Proposition 3.1

Given the definition in 3.1, the evaluation bias on benchmark $D$ is given by:

$$\epsilon(\hat{\phi}_D) = \mathbb{E}[\hat{\phi}_D] - \phi \qquad (8)$$

Similarly, the evaluation bias on rephased one $D'$ from $D$ is defined as $\epsilon_{D'}$, then

$$
\begin{aligned}
\epsilon(\hat{\phi}_{D'}) &= \mathbb{E}[\hat{\phi}_{D'}] - \phi \\
&= \mathbb{E}[\hat{\phi}_{D'}] - (\mathbb{E}[\hat{\phi}_D] - \epsilon(\hat{\phi}_D)) \\
&= (\mathbb{E}[\hat{\phi}_{D'}] - \mathbb{E}[\hat{\phi}_D]) + \epsilon(\hat{\phi}_D) \\
&= \Delta + \epsilon(\hat{\phi}_D)
\end{aligned}
\tag{9}
$$

where $\Delta$ is the delta bias which arises from the introduction of new evaluation protocol.

Consider that $\epsilon(\hat{\phi}_{D'})$ could be positive, negative, and ideally, zero. Therefore, we seek to analyze the its mean squared term:

$$
\begin{aligned}
\mathbb{E}[\epsilon(\hat{\phi}_{D'})^2] &= \mathbb{E}[(\Delta + \epsilon(\hat{\phi}_D))^2] \\
&= \mathbb{E}[\Delta^2] + \mathbb{E}[\epsilon(\hat{\phi}_D)^2] + 2\mathbb{E}[\Delta\epsilon(\hat{\phi}_D)]
\end{aligned}
\tag{10}
$$

Because,

$$
\begin{aligned}
\mathrm{Cov}(\epsilon(\hat{\phi}_D), \Delta) &= \mathbb{E}[(\epsilon(\hat{\phi}_D) - \mathbb{E}[\epsilon(\hat{\phi}_D)])(\Delta - \mathbb{E}[\Delta])] \\
&= \mathbb{E}[\epsilon(\hat{\phi}_D)\Delta - \epsilon(\hat{\phi}_D)\mathbb{E}[\Delta] - \Delta\mathbb{E}[\epsilon(\hat{\phi}_D)] + \mathbb{E}[\epsilon(\hat{\phi}_D)]\mathbb{E}[\Delta]] \\
&= \mathbb{E}[\epsilon(\hat{\phi}_D)\Delta] - \mathbb{E}[\epsilon(\hat{\phi}_D)]\mathbb{E}[\Delta] - \mathbb{E}[\Delta]\mathbb{E}[\epsilon(\hat{\phi}_D)] + \mathbb{E}[\epsilon(\hat{\phi}_D)]\mathbb{E}[\Delta] \\
&= \mathbb{E}[\epsilon(\hat{\phi}_D)\Delta] - \mathbb{E}[\epsilon(\hat{\phi}_D)]\mathbb{E}[\Delta]
\end{aligned}
\tag{11}
$$

Therefore,

$$
\mathbb{E}[\epsilon(\hat{\phi}_D)\Delta] = \mathrm{Cov}(\epsilon(\hat{\phi}_D), \Delta) + \mathbb{E}[\epsilon(\hat{\phi}_D)]\mathbb{E}[\Delta]
\tag{12}
$$

Substitute the expression into Equation (10),

$$
\mathbb{E}[\epsilon(\hat{\phi}_{D'})^2] = \mathbb{E}[\epsilon(\hat{\phi}_D)^2] + 2\mathrm{Cov}(\epsilon(\hat{\phi}_D), \Delta) + 2\mathbb{E}[\epsilon(\hat{\phi}_D)]\mathbb{E}[\Delta] + \mathbb{E}[\Delta^2]
\tag{13}
$$

Here,

- $\mathbb{E}[\epsilon(\hat{\phi}_D)^2]$ is an original term that is associated with the original bias existing in the original benchmark $D$.

- $2\mathrm{Cov}(\epsilon(\hat{\phi}_D), \Delta)$ is a related term which pertains to biases that newly introduced biases are correlated with the pre-existing biases in the original benchmark.

- $2\mathbb{E}[\epsilon(\hat{\phi}_D)]\mathbb{E}[\Delta] + \mathbb{E}[\Delta^2]$ is an independent term, stemming from biases inherent to the methodology itself.

## E. Intervened Examples

```
Here is a multiple choice question. If there is no correct answer in the options, please
    reply with N.
Question:
The end result in the process of photosynthesis is the production of sugar and oxygen.
    Which step signals the beginning of photosynthesis?
I. Chemical energy is absorbed through the roots.
II. Light energy is converted to chemical energy.
III. Chlorophyll in the leaf captures light energy.
IV. Sunlight is converted into chlorophyll.
Answer:
III
```

Example #1

```
Here are two questions and only one of them corresponds to the options. Please select the
    correct answer.
Question:
```

A group of engineers wanted to know how different building designs would respond during an
    earthquake. They made several models of buildings and tested each for its ability to
    withstand earthquake conditions. Which will most likely result from testing different
    building designs?
The voltage is held constant in an electric circuit. What will happen to the current in
    this circuit if the resistance is doubled?
A. buildings will be built faster
B. buildings will be made safer
C. building designs will look nicer
D. building materials will be cheaper
Answer:
B

## Example #2

Here are two questions and only one of them corresponds to the options. Please select the
    correct answer. If there is no correct answer in the options, please reply with N.
Question:
Which statement best explains why a tree branch floats on water?
What happens to a wooden log when it is burned?
I. Wood is light.
II. parasitism
III. Wood is magnetic.
IV. Wood is porous.
Answer:
N

## Example #3

The following are true/false questions. If the answer is correct, please reply with T,
    otherwise reply with F.
Question:
Statement 1: A polished metal ball looks very shiny and bright on a sunny day. What makes
    the ball look shiny? The ball makes light.
Statement 2: A polished metal ball looks very shiny and bright on a sunny day. What makes
    the ball look shiny? The ball reflects light.
Statement 3: A polished metal ball looks very shiny and bright on a sunny day. What makes
    the ball look shiny? The ball absorbs light and then releases it.
Statement 4: A polished metal ball looks very shiny and bright on a sunny day. What makes
    the ball look shiny? The ball absorbs light and keeps it inside.
Answer:
F T F F

## Example #4

