# OpenReview forum: "Unbiased Evaluation of Large Language Models from a Causal Perspective"
_ICML.cc/2025/Conference — ICML 2025 poster_

### Official Review · Reviewer_zS8E · 2025-03-13

**Overall Recommendation:** 4

**Summary:**

The paper explores bias in agents as evaluators (LLMs generating new tasks for evaluating another agent), and detects different kinds of biases. They introduce an unbiased evaluator using causal inference.

"## update after rebuttal"
I thank the authors for answering my questions. After seeing the other reviews and discussion, I still have doubts about the significance of the paper, but some other issues have disappeared so I increase the score.

**Claims And Evidence:**

- The problem of bias is very important when using LLMs evaluating other LLMs.
- The causal interventions are not based on the causal graph, even if the causal graph was introduced to understand the bias.
- The interventions may overcompensate? It's not clear the reduction of performance is actually (only) compensating for the bias.

**Essential References Not Discussed:**

No

**Experimental Designs Or Analyses:**

Yes, I checked the design as written in the paper.

**Methods And Evaluation Criteria:**

Yes.

**Other Comments Or Suggestions:**

Reference AI-MO 2024 should be AIME 2024?

**Other Strengths And Weaknesses:**

No

**Questions For Authors:**

Table 2 is hard to understand and going to appendix D didn't help in my case. In Figure 3 the options and the option of "no correct" is introduced, but what's the problem with the original questions, other than contamination?

And the transformation makes them more difficult, but perhaps not only because of removing contamination. I don't see a clear causal graph justifying the intervention. Can these two things being separated?


The baseline not changing the meaning is an interesting baseline, but where is this used?


Figure 5 shows the accuracy goes down, but I'm not sure that means that contamination goes down as well. How can we know?

**Relation To Broader Scientific Literature:**

I miss an independent assessment of question difficulty by humans, to understand if the interventions are changing some other things. The reformulation should give exactly the same results for humans, and some others that make the questions more difficult should create a similar effect as in humans.
For the role of difficulty: Adversarial Benchmark Evaluation Rectified by Controlling for Difficulty
https://www.researchgate.net/publication/374304817_Adversarial_Benchmark_Evaluation_Rectified_by_Controlling_for_Difficulty

**Theoretical Claims:**

The theoretical decomposition. I haven’t checked it and find it a bit too abstract, and perhaps not that relevant.

---

> ### Author Rebuttal · Authors · 2025-03-31
>
> ## Response to reviewer zS8E
> > **Q1**: Reference AI-MO 2024 should be AIME 2024
>
> **A1:** AI-MO (refer to https://aimoprize.com), the AI Mathematical Olympiad, adapts data from AIME 2024 as its competition benchmark. This widely-used version is publicly available as [aimo-validation-aime](https://huggingface.co/datasets/AI-MO/aimo-validation-aime). Therefore, as referenced in L474, we cite it (in-text citation appears as AI-MO, 2024.) as:
> ```
> AI-MO. AIME 2024. https://huggingface.co/datasets/AI-MO/aimo-validation-aime, 2024.
> ```
>
>
> > **Q2**: Table 2 is hard to understand and going to appendix D didn't help in my case.
>
> **A2:** Table 2 illustrates a demo case of Bags of Atomic Interventions (BOAT) to demonstrate how each intervention works. For each intervention, we present a simple question to show its effect. To save space, the "intervened" column only displays the part that has changed, highlighted in the same color as the original part it replaces. The other parts of the content remain unchanged. For example, with the Distractor Hint intervention, the full modified content would look like this:
> ```
> ## original question
> Question: Here is a multiple choice question, answer A or B. Is 9.8 bigger than 9.11?
> Option: A: True B: False
> Label: A
>
> ## intervened question
> Question: Here is a multiple choice question, answer A or B, if there is no answer, reply N. Is 9.8 bigger than 9.11?
> Option: A: True B: False
> Label: A
> ```
> We will add detailed demonstrations into the caption of Table 2 for next version.
>
> > **Q3**: In Figure 3 the options and the option of "no correct" is introduced, but what's the problem with the original questions, other than contamination?
>
> **A3:** As detailed in our general evaluation formulation (L253), we argue that the evaluation process can be seen as a causal analysis. It is essential to assess whether the model truly understands and can make these causal connections. In this context, the Distractor Hint/Answer Removal is designed to evaluate not only whether the model selects the correct answer, but also whether it effectively rejects the incorrect options.
>
> > **Q4**: The transformation makes them more difficult, but perhaps not only because of removing contamination. Can these two things being separated?
>
> **A4:** We argue that our Unbiased Evaluator does **NOT** inherently make questions more difficult. The adversarial benchmark in paper [1] mentioned by the reviewer is fundamentally different from our approach. Specifically, adversarial benchmarks are designed to exploit a model’s weaknesses, often **using supervised optimization algorithms to find desired perturbations** that lead to incorrect predictions. In contrast, our Unbiased Evaluator aims to assess **whether models can genuinely answer a question correctly by employing causal interventions that align with human recognition**. Therefore, rather than increasing difficulty, our Unbiased Evaluator provides a more accurate measure of a model’s true and robust performance on a given benchmark by eliminating performance inflation caused by data contamination.
>
> To clarify the functioning of the Unbiased Evaluator, we have included a separation ablation in Figure 6, which demonstrates the impact of each individual intervention. Even simple manipulations, such as Option Shuffling, Label Replacement, and Binary Transformation, lead to noticeable degradation in the model’s performance. This provides strong evidence that data contamination plays a significant role in inflating evaluation outcomes.
>
> > **Q5**: The baseline not changing the meaning is an interesting baseline, but where is this used?
>
> **A5:** The rephasing baseline, without changing the question's meaning, is referred to as a minimal Agents-as-an-Evaluator in this paper (see L208), and it serves as a comparative baseline in Figure 2 and Table 1.
>
> > **Q6**: Figure 5 shows the accuracy goes down, but I'm not sure that means that contamination goes down as well. How can we know?
>
> **A6:** As presented in A4, Unbiased Evaluator evaluates a model's true and robust performance on a given benchmark. Therefore, **the decline of accuracy actually reflects the decrease of contamination**. To validate this, we further provide an additional fine-tuning ablation study. Specifically, we fine-tune Llama2-13B on the original samples from the MMLU test set and evaluate it on MMLU test set under two conditions: with and without our Unbiased Evaluator.
>
> |train set|w/o Unbiased Evaluator|w/ Unbiased Evaluator|
> |----------------------|---------|---------|
> |Llama2-13B|55.6|33.7|
> |Llama2-13B + original test set|96.6|37.1|
>
> Even when trained directly on the original test set, the model struggles to perform well under the Unbiased Evaluator, suggesting that it effectively mitigates data contamination and ensures a more robust evaluation.
>
> [1] Adversarial benchmark evaluation rectified by controlling for difficulty

---

### Official Review · Reviewer_A1m7 · 2025-03-14

**Overall Recommendation:** 3

**Summary:**

This paper studies potential biases in LLM-based evaluators (“Agents-as-an-Evaluator”) and proposes a new protocol, called the “Unbiased Evaluator,” which systematically introduces small interventions (“Bags Of Atomic Interventions”) into evaluation tasks to mitigate data and model biases. The authors present both theoretical and empirical analyses suggesting their protocol reduces correlation based artifacts and helps reveal model weaknesses that standard benchmarks may overlook.

**Claims And Evidence:**

The main claim is that existing multi agent evaluators introduce bias during question generation, and that the proposed BOAT based evaluator offers a more “unbiased” alternative. While the experiments (notably the confusion matrices) do highlight differences in evaluation outcomes, the evidence for truly mitigating all bias remains somewhat limited and relies on a relatively small set of carefully selected interventions. It is unclear whether these interventions comprehensively address the broad range of biases in LLM based evaluations.

**Essential References Not Discussed:**

I think mentioning how common bias mitigation approaches proposed within the LLM-as-a-Judge framework is related to the paper, such as Length-controlled AlpacaEval (Dubois et al) and Arena-Hard Style Control (Li et al).

**Experimental Designs Or Analyses:**

The experiments cover multiple model sizes (both open-source and proprietary), several well-known benchmarks (ARC, MMLU, GSM8K), and detailed ablations of single versus combined interventions.

Human verification of a subset of transformed samples is used to confirm correctness of the approach (high agreement rate).

The methods are transparent, and the sample sizes are standard for these benchmarks, though additional clarity on how random interventions might differ across runs could improve reproducibility.

**Methods And Evaluation Criteria:**

The authors conduct a systematic causal analysis, framing QA as a DAG with interventions on specific “atomic” components.

They define several carefully controlled transformations, such as adding distractor questions, to stress test LLM understanding.

Accuracy across these perturbed scenarios is aggregated and compared with standard benchmarks.

**Other Comments Or Suggestions:**

N/A

**Other Strengths And Weaknesses:**

**Strengths**:
1. Presents a fresh causal perspective on LLM evaluations.
2. Detailed metrics for identifying overconfidence and underconfidence biases.
3. Scalability: the method can be adapted to various choice-based tasks.

**Weaknesses**:
1. Some aspects of the theoretical framework remain high-level; more rigorous proofs or formal constraints on interventions might strengthen the argument.
2. The paper focuses primarily on multiple-choice formats; it would be insightful to see how the method generalizes to more open-ended tasks.

**Questions For Authors:**

1. How would the proposed atomic interventions scale to more complex structured tasks beyond multiple-choice towards open-ended prompt?
2. Did you observe any qualitative differences in model performance across different types of math or reasoning questions when interventions stack up?
3. Can you provide more details on how changes in model performance under your method correlate with human expert judgments overall?

**Relation To Broader Scientific Literature:**

The paper builds on existing work on LLM-as-a-Judge by extending to “Agents-as-an-Evaluator" by dissecting biases in both question generation and model self-assessment. Drawing on causal-inference ideas (e.g., interventions on input variables), it aligns with literature on benchmark contamination and fairness in NLP.

**Theoretical Claims:**

The decomposition of evaluation bias appears logically consistent, and the provided proofs in the appendix are straightforward, though the argument is more conceptual than heavily formal.

---

> ### Author Rebuttal · Authors · 2025-03-31
>
> ## Response to reviewer A1m7
> > **Q1**: Bias mitigation approaches in the LLM-as-a-Judge are related to the paper, such as Length-controlled AlpacaEval and Arena-Hard Style Control.
>
> **A1:** Thank you for your suggestion. Unlike previous bias mitigation approaches in the LLM-as-a-Judge, which primarily focus on the judge side, our paper is the first to analyze bias from the generation side, i.e., Agents-as-an-Evaluator. We will incorporate these works into our related work section for clarification.
>
> > **Q2**: Some aspects of the theoretical framework remain high-level, and more rigorous proofs or formal constraints on interventions might strengthen the argument.
>
> **A2**: We argue that **our theoretical framework is intuitive and sufficient to support our method’s design**. Inspired by findings in Proposition 3.1, BOAT is designed to mitigate the impact of the related and independent terms (detailed in L366). Further refinements and formal extensions of our framework are important directions that we plan to explore in future work.
>
> > **Q3**: How would the proposed atomic interventions scale to more complex structured tasks beyond multiple-choice towards open-ended prompt?
>
> **A3:** As shown in Table 3, our method, with designed Question and Answer Jitter, has been successfully scaled to the mathematics benchmark GSM8K, which does not rely on multiple-choice style. Based on our causal formulation of evaluation, we can categorize tasks into two types.
> - **Most tasks** (e.g., multiple-choice, math), inherently follow natural rules in either the questions or answers, and rule-based interventions can be automatically applied.
> - **The other small percentage of tasks** (e.g., CivilComments), can use a **debiased Agents-as-Evaluator version**. Concretely, our study has revealed the data and model biases of previous version, inspiring two designs to mitigate them: (1) cross-generation: to reduce model bias, we can break down question generation into multiple chunks, using different models for each. (2) cross-checking: multiple advanced models can be used to cross-check the output to mitigate data bias and enhance quality.
>
> Overall, our method is easily scaled to most tasks, and our insights will provide valuable inspiration for future advancements in evaluation methodologies. We will add a section of **Future Work** to include these discussions.
>
> > **Q4**: Did you observe any qualitative differences in model performance across different types of math or reasoning questions when interventions stack up?
>
> **A4:** Yes. In addition to GSM8K in Table 3, we conduct further experiments on a more challenging benchmark, MATH500. We also evaluate a recently open-sourced reasoning model, QWQ-32B, with 16k context. Our experiments revealed two key observations.
>
> - **the performance gap between models becomes more pronounced from GSM8K to MATH500**. Notably, Qwen2.5-72B remains the strongest, on par with Mistral-Large-2411 (123B). Meanwhile, the gap between Qwen2.5-72B and models like Llama3.1-70B widens considerably—rising from 5.89 on GSM8K to 19.56 on MATH500, highlighting the superior capabilities of Qwen2.5-72B and Mistral-Large-2411 in handling complex mathematical reasoning.
> - **the reasoning model exhibits stronger generalization on mathematical benchmarks**, experiencing a significantly smaller performance drop compared to others.
>
> |Model|GSM8K Vanilla|GSM8K Ours|Δ|MATH500 Vanilla|MATH500 Ours|Δ|
> |-|-|-|-|-|-|-|
> |Qwen2.5-72B|98.41|88.86|9.55|92.23|77.57|14.66|
> |Llama3.1-70B|95.98|82.97|13.01|75.87|58.01|17.86|
> |Yi1.5-34B|91.96|69.60|22.36|65.44|56.37|9.07|
> |Mistral-Large-2411|97.73|90.04|7.69|86.71|77.51|9.20|
> |QWQ-32B(16k)|99.32|95.32|4.00|89.78|88.18|1.60|
>
> > **Q5**: How changes in model performance under your method correlate with human expert judgments overall?
>
> **A5:** **Our Unbiased Evaluator provides a much more correlative assessment with human expert judgments**.
> Since collecting overall expert judgments across multiple model is costly and impractical, we instead compare our method with LiveBench, a continuously updated benchmark. Specifically, we compute the Pearson and Kendall correlations between our averaged results (Table 3) and the global average results in latest LiveBench-2024-11-25 (https://livebench.ai). Notably, we exclude two models (GPT-4-Turbo and Yi1.5-34B-Chat) that are not evaluated in LiveBench-2024-11-25 for a fair comparison.
>
> ||Pearson|Kendall|
> |-|-|-|
> |Vanilla|0.918|0.600|
> |Unbiased Evaluator|0.949|1.000|
>
> These results confirm that our method aligns more closely with LiveBench. Notably, it achieves a perfect ranking correlation with LiveBench (as measured by Kendall), a significant improvement over baseline. Unlike LiveBench, which covers diverse tasks and requires substantial resources to update questions regularly, ours leverages existing benchmarks and requires almost no additional resources. We sincerely appreciate your valuable suggestions and will add these results to our ablations.

---

### Official Review · Reviewer_3rKe · 2025-03-17

**Overall Recommendation:** 4

**Summary:**

The paper introduces ‘Agent as an evaluator’ paradigm with the goal to increase the robustness of LLM-as-a-Judge based evaluations. The evaluation protocol introduces the ability to test model and data bias by taking an active/intervening (agentic) process of evaluating the benchmarks. The query breakdown focuses on problem rephrasing to assess stability of responses. The authors design probing tasks to identify various contamination effect biases. These tasks are designed to reveal data and model biases, informing the development of the Unbiased Evaluator. The work is somewhat inspired from similar research like CogMath where CogMath formalizes the reasoning process into three stages: problem comprehension, problem solving, and solution summarization - however this research generalizes to all other of evaluations as well as introduces error breakdown and analysis with theoretical underpinning and shows strong correlations with performance metrics using statistical metrics.

**Claims And Evidence:**

The contributions (according to authors) are as follows:
- A theoretical formulation of evaluation bias, offering valuable findings for the importance of minimizing the relative term when designing evaluation protocols.
- The first comprehensive bias analysis for Agents-as-an-Evaluator, revealing data and model bias which undermine the reliability and trustworthiness of Agents-as-an-Evaluator.
- An unbiased evaluation protocol, Unbiased Evaluator, provides a more comprehensive, unbiased and interpretable assessment for benchmark contamination.

The claims are backed with proofs, design of experiments and various results and analysis to validate the claims. There are some similar work that the authors have attributed to in this paper.

**Essential References Not Discussed:**

Seems pretty good. However, I may have missed some theoretical references related to studying various error types.

**Experimental Designs Or Analyses:**

The experiments make a lot of sense validating the experimental design and analysis.

**Methods And Evaluation Criteria:**

The Unbiased Evaluator employs 'BOAT' based probing method to dynamically assess LLMs, aiming to reduce evaluation biases (unlike the baseline evaluation that the authors say give an unfair advantage to larger LLMs that show higher over-confidence, for example). This method seeks to provide a more accurate representation of an LLM’s capabilities by minimizing the influence of data and model biases.

**Other Comments Or Suggestions:**

Agents-as-an-Evaluator seems like a new and slightly confusing term - needs some definition and clarification imo.

**Other Strengths And Weaknesses:**

- The paper writing and organization can be improved a lot. The paper starts and shows the cryptic Fig. 1 and non-standard Fig 2 (and talks about Fig 2 much later) - the definitions are vague (still confused how strength parameter is varies during evaluations)
- There is very little (almost no) comparison to other work in the field. For example, authors refer to Ye et al. 2024 work ( https://arxiv.org/pdf/2410.0273) where they refer to various biases coming from LLM-as-a-Judge and how the work has defined Robustness rate and consistency rate metrics to assess some of these biases - how do these compare with this work or other relevant work?

**Questions For Authors:**

- please explain the strength parameter and it is varied in the experiments

**Relation To Broader Scientific Literature:**

This work generalizes evaluations using LLM-as-a-Judge paradigm which is one of the few scalable methods today for LLM evaluations (without humans in the loop). It decomposes the evaluation by breaking up the generation, coming up with various probes and then defining theoretical framework for measuring various metrics (consensus, OC, UC).
This field is emerging where currently there is a lack of rigor in evaluation for most benchmarks.

**Theoretical Claims:**

The unbiased evaluation protocol systematically applies statistical principles to decompose the evaluation bias. The decomposition into original, related, and independent terms provides valuable insights into how new biases (using probes) interact with existing ones, guiding the design of more unbiased evaluation protocols.

---

> ### Author Rebuttal · Authors · 2025-03-31
>
> ## Response to reviewer 3rKe
> > **Q1**: The paper writing and organization can be improved a lot. The paper starts and shows the cryptic Fig. 1 and non-standard Fig 2 (and talks about Fig 2 much later) - the definitions are vague (still confused how strength parameter is varies during evaluations)
>
> **A1-part1 (for Figure 1):**
> Figure 1 illustrates the overall pipeline of Agents-as-an-Evaluator and our proposed Unbiased Evaluator.
> The Agents-as-an-Evaluator process (Figure 1a) consists of generation and evaluation stages. The generation phase is affected by data bias (Table 1), where LLMs tend to perform generation task (such as rephrasing) significantly worse in domains where their evaluation performance are weaker. Furthermore, the evaluation phase involves model bias — LLMs generate content that aligns more closely with their strengths, giving themselves an unfair advantage (represented by term "familiar"). In contrast, our proposed Unbiased Evaluator (Figure 1b) evaluates the LLMs with designed BOAT.
> Considering that Figure 1 may lead to potential misinterpretation, **we provide a simplified version (refer to https://i.imgur.com/XpnwLsk.jpeg)**. We will update Figure 1 with this version and provide a more detailed caption to enhance clarity.
>
>
> **A1-part2 (for Figure 2 and strength parameter):**
> Figure 2 visualizes the variation of two proposed metrics ($R_{OC}$ on the left and $R_{UC}$ on the right) as strength parameter changes, using two datasets (MMLU and ARC-C). As stated in L267, **strength refers to the probability defined in Equation 3. A higher strength value indicates a greater proportion of "processed" samples within the dataset** ("process" denotes rephrasing and BOAT in Agents-as-an-Evaluator and Unbiased Evaluator, respectively).
> As the strength increases, for Agents-as-an-Evaluator, we observe a significant rise in $R_{OC}$, while $R_{UC}$ remains relatively stable, suggesting the existence of model bias. In contrast, our Unbiased Evaluator remains relatively stable on both metric.
>
> We will incorporate this clarification of the strength parameter into the caption and relocate Figure 2 closer to Section 3.3 for better alignment.
>
> > **Q2**: There is very little (almost no) comparison to other work in the field. For example, authors refer to Ye et al. 2024 work ( https://arxiv.org/pdf/2410.02736) where they refer to various biases coming from LLM-as-a-Judge and how the work has defined Robustness rate and consistency rate metrics to assess some of these biases - how do these compare with this work or other relevant work?
>
> **A2**: As demonstrated in A1, we **DO** compare our method with previous relevant works on two widely-used benchmarks (MMLU and ARC-C), considering both types of bias. Building upon the theoretical findings and the first comprehensive bias analysis of Agents-as-an-Evaluator, our method is designed as an unbiased LLM evaluation protocol. Therefore, we compare Unbiased Evaluator with previous Agents-as-an-Evaluator on both data (Table 1) and model bias (Figure 2).
>
> As for previous evaluation bias works, such as [Ye et al. 2024]( https://arxiv.org/pdf/2410.02736), we have presented detailed discussion on Section 2.3 and L197. Prior works mainly focus on biases in LLM-as-a-Judge, which operates on the judge side by solely determining whether input falls within the scope of a given rule (e.g. score range 0~5). In contrast, our paper is the first to address the biases inherent in the generation side of Agents-as-an-Evaluator, where LLMs actively contribute to the generation of the very questions.
>
> > **Q3**: "Agents-as-an-Evaluator" seems like a new and slightly confusing term and needs some definition
>
> **A3**: Integrating agents into the evaluation process is a very recent research direction. Building on the concept of LLM-as-a-Judge, we introduce the term Agents-as-an-Evaluator and have clarified its distinction from LLM-as-a-Judge in L48-L53. Formally, Agents-as-an-Evaluator refers to an LLM-based evaluation paradigm in which LLMs (or Agents) not only assess responses but also actively contribute to generating evaluation criteria and questions. We will incorporate this formal definition into the introduction for better clarity.

---

### Official Review · Reviewer_D15D · 2025-03-25

**Overall Recommendation:** 3

**Summary:**

This paper presents Bags of atomic interventions (BOAT) to address the data contamination problem in LLM evaluation. It first develops a theoretical formulation of evaluation bias, and identity the data and model bias in agents-as-an-evaluator paradigm. It then proposes the unbiased evaluator to help evaluate LLMs with less bias.

## update after rebuttal
In my initial comment, I mainly question the justification of BOAT. During the rebuttal, the author has thoroughly addressed this concern, so I updated my score to support the work.

**Claims And Evidence:**

Yes or no.

One of the major claim in the paper is supported by Table 3 which indicates the contamination problem in the current benchmark. The unbiased evaluator heavily depends on the BOAT, which is hand-designed (Section 4.2). The reviewer is not fully aware of how these principles are hand-designed to fully follow the theoretical framework.

**Essential References Not Discussed:**

Please see above comments.

**Experimental Designs Or Analyses:**

The experimental designs largely make sense, but the reviewer is not convinced of the derivation of BOAT.

**Methods And Evaluation Criteria:**

Yes, the paper uses ARC-C and MMLU, GSM8K for evaluation, gpt-4 and gemini, llama, mistral, qwen and yi for models. The reviewer is convinced these choices are reasonable.

**Other Comments Or Suggestions:**

Please see the above comments.

**Other Strengths And Weaknesses:**

Other strength:
 The paper is addressing an important problem and attempt from a theoretical perspective. The major weakness is the justification of BOAT and does not distinguish against the above paper. The reviewer is willing to raise the score they can be addressed adequately.

**Questions For Authors:**

Please see above questions.

**Relation To Broader Scientific Literature:**

The paper largely cited proper papers. However, the reviewer believes there is a popular and similar work that the paper does not discuss [1]. Can the author includes the discussion against this paper?

[1] Rethinking Benchmark and Contamination for Language Models with Rephrased Samples.

**Theoretical Claims:**

Yes, the reviewer checks the theoretical analysis in Section 3.

---

> ### Author Rebuttal · Authors · 2025-03-31
>
> ## Response to reviewer D15D
> > **Q1**: The unbiased evaluator heavily depends on the BOAT, which is hand-designed, and how these principles are hand-designed to fully follow the theoretical framework.
>
> **A1**: **The design of the Unbiased Evaluator is grounded in our theoretical findings** (see the detailed discussion in L366). In particular, Proposition 3.1 shows that the bias in the new evaluation protocol can be decomposed into original, related, and independent terms. For the related term, BOAT’s interventions help mitigate the biases present in the original benchmark (such as ambiguities), thus reducing the impact of the related term in Proposition 3.1. Additionally, the independent term is minimized by our rule-based design.
>
> Overall, this paper provides the first comprehensive bias analysis for Agents-as-an-Evaluator and designs a simple unbiased alternative guided by our theoretical insights. Our theoretical insights, as well as bias analysis, will inspire future design for LLM evaluation.
>
>
> > **Q2**: Discussion with previous paper [1]
>
> **A2**: We argue that our paper differs from paper [1] in the following aspects:
>
> - **Different focus, findings, and methodologies**: First, paper [1] mainly addresses **contamination**, specifically the inclusion of rephrased test samples in training data. In contrast, our work focuses on **evaluation bias** in the Agents-as-an-Evaluator paradigm (rephrasing is a special case of Agents-as-an-Evaluator). Second, while paper [1] utilizes an LLM-based decontaminator to identify rephrased samples, we take a fundamentally different approach by mitigating evaluation bias through causal interventions.
>
> - **Our method naturally extends and advances the contributions of paper [1]:** Paper [1] highlighted the challenges in contamination formulation for future works in the end (see Section 6.1), such as mathematical cases where a training and test example differ only in numerical values and background details. As outlined in our general evaluation formulation (L253), we formulate the evaluation process into a causal analysis, and it is crucial to assess whether the model is genuinely capable of these causal connections. Based on this, a robust and contamination-free evaluation protocol should determine whether the model truly possesses the ability to answer the questions correctly. Our proposed Unbiased Evaluator achieves this by assessing the model’s responses under various causal combinations of Bags of Atomic Interventions (BOAT).
>
> For a more comprehensive understanding of the Unbiased Evaluator, following the contamination detection methodology in [1], we perform an evaluation of the fine-tuned model using our approach. Specifically, we fine-tune Llama2-13B on both rephrased and original samples from the MMLU test set and evaluate it on MMLU test set under two conditions: with and without our Unbiased Evaluator. In particular, the results in parentheses are the results from Table 2 of paper [1].
>
> |train set|w/o Unbiased Evaluator|w/ Unbiased Evaluator|
> |----------------------|---------|---------|
> |Llama2-13B|55.6 (54.8)|33.7|
> |Llama2-13B + rephrased test set|85.7 (85.9)|32.8|
> |Llama2-13B + original test set|96.6 (100)|37.1|
>
> These results highlight that our Unbiased Evaluator provides a more rigorous assessment of benchmark contamination. Even when trained directly on the original test set, the model struggles to perform well under the Unbiased Evaluator, suggesting that it effectively mitigates data contamination and ensures a more robust evaluation.
>
> Overall, grounded in our theoretical findings and the first bias analysis for Agents-as-an-Evaluator, Unbiased Evaluator is designed to provide a more robust and unbiased assessment for benchmark contamination. We sincerely appreciate your valuable suggestions and will cite paper [1] and include the discussions above into the related works section.
>
> If our rebuttal successfully addresses your concerns, we kindly ask you to consider raising our score.
>
> [1] Rethinking Benchmark and Contamination for Language Models with Rephrased Samples

---

> > ### Comment · Reviewer_D15D · 2025-04-02
> >
> > Thank you for getting back. I appreciate the rebuttal; they address my concerns. I raise my score to 3 to support the paper.

---

### Decision · Program_Chairs · 2025-05-01

**Decision:**

Accept (poster)

**Comment:**

The paper provides a causal framework to evaluate agents as an evaluator LLMs. The paper provides an evaluation protocol by designing a handful of rule‑based edits (BOAT) such as option shuffling, distractor insertion and label replacement to mitigate different sources of bias. It attempts to formally motivate these interventions via a decomposition of the evaluator bias. Empirically, they apply the protocol to different standard LLM benchmarks and show that it can mitigate inflated scores and finds close agreement of the model rankings with livebench. Another ablation shows that finetuning on the test set directly would yield substantially lower gains for models under the proposed evaluation protocol providing further evidence to support the key claims.

There are limitations to the proposed approach: the formalism is high level and the proposed interventions are hand crafted and are not naturally generalizable say to more open ended QA benchmarks. Furthermore the proposed interventions could be modelled as augmentations to the training dataset limiting the generalizability of the specific interventions.

Nevertheless, all reviewers agree that the paper addresses a timely issue: trustworthy evaluation of LLMs and the proposed protocol gives a pragmatic, well‑validated solution. Initial concerns about the hand‑crafted nature of BOAT and the clarity of figures seemed to have been somewhat clarified in the rebuttal. After clarifications and new experiments, all reviewers lean to accept; the method’s practicality and breadth outweigh the aforementioned limitations so I recommend acceptance.